# Dihydrotanshinone I Enhances Cell Adhesion and Inhibits Cell Migration in Osteosarcoma U−2 OS Cells through CD44 and Chemokine Signaling

**DOI:** 10.3390/molecules27123714

**Published:** 2022-06-09

**Authors:** Lanyan Fan, Chen Peng, Xiaoping Zhu, Yawen Liang, Tianyi Xu, Peng Xu, Shihua Wu

**Affiliations:** 1Joint Research Centre for Engineering Biology, Zhejiang University-University of Edinburgh Institute, Zhejiang University, Haining 314400, China; 21707013@zju.edu.cn (L.F.); zhuxiaoping@zju.edu.cn (X.Z.); xupeng126@zju.edu.cn (P.X.); 2Research Center of Siyuan Natural Pharmacy and Biotoxicology, College of Life Sciences, Zhejiang University, Hangzhou 310058, China; 3180105399@zju.edu.cn (C.P.); 22007006@zju.edu.cn (Y.L.); 21807009@zju.edu.cn (T.X.)

**Keywords:** CD44, cell adhesion, cell migration, dihydrotanshinone I, osteosarcoma, *Salvia miltiorrhiza* Bunge (Danshen)

## Abstract

In the screening of novel natural products against cancer using an in vitro cancer cell model, we recently found that tanshinones from a traditional Chinese medicine, the rhizome of *Salvia miltiorrhiza* Bunge (Danshen), had potent effects on cell proliferation and migration. Especially for human osteosarcoma U−2 OS cells, tanshinones significantly enhanced the cell adherence, implying a possible role in cell adhesion and cell migration inhibition. In this work, therefore, we aimed to provide a new insight into the possible molecule mechanisms of dihydrotanshinone I, which had the strongest effects on cell adhesion among several candidate tanshinones. RNA−sequencing-based transcriptome analysis and several biochemical experiments indicated that there were comprehensive signals involved in dihydrotanshinone I-treated U−2 OS cells, such as cell cycle, DNA replication, thermogenesis, tight junction, oxidative phosphorylation, adherens junction, and focal adhesion. First, dihydrotanshinone I could potently inhibit cell proliferation and induce cell cycle arrest in the G0/G1 phase by downregulating the expression of *CDK4*, *CDK2*, *cyclin D1*, and cyclin E1 and upregulating the expression of p21. Second, it could significantly enhance cell adhesion on cell plates and inhibit cell migration, involving the hyaluronan CD44−mediated CXCL8–PI3K/AKT–FOXO1, IL6–STAT3–P53, and EMT signaling pathways. Thus, the increased expression of CD44 and lengthened protrusions around the cell yielded a significant increase in cell adhesion. In summary, these results suggest that dihydrotanshinone I might be an interesting molecular therapy for enhancing human osteosarcoma U−2 OS cell adhesion and inhibiting cell migration and proliferation.

## 1. Introduction

Cancer ranks as a leading cause of death and an important barrier to increasing life expectancy in every country of the world [1]. For example, in children and adolescents, osteosarcoma is the most common primary malignant bone tumor [2,3]. There are many advances that have been achieved in the biological understanding of cancers. However, the progress made in improving the overall survival in patients with osteosarcoma is still limited, owing to its strong ability related to invasion, metastasis, and rapid proliferation [4,5,6,7]. Therefore, how to effectively inhibit its growth and control its metastasis maintains a hot issue in current drug research. It is necessary to develop new molecular therapies for this metastatic bone cancer.

Natural products are still thought of as important resources of drug discovery and pharmaceutical agents due to their diversity in structure and bioactivities [8,9]. During the past nearly 40 years, about half of all FDA-approved drugs were derived from natural products, especially for anti-cancer and anti-inflammatory drugs [9,10]. This still seems a promising way to discover efficient and novel natural products against invasive osteosarcoma in the genomics age [11,12]. Using several cancer cells including typical human osteosarcoma U−2 OS cells [13,14], classical human cervical carcinoma HeLa cells, and normal rat kidney cells NRK−49F as in vitro cell models, recently, we set out to comprehensively screen active natural products from versatile natural resources, including herbal plants and macromycetes (fungi). We found that there were a large number of natural products possessing potent cytotoxic activity against U−2 OS cells, such as taxoids, celastrols, podophyllotoxins, cytochalasins, and piper amide alkaloids. Interestingly, we found that tanshinones, especially dihydrotanshinone I, possessed potent activity to enhance U−2 OS cell adhesion on cell plates and inhibited cell migration.

As shown in Figure 1 and Appendix A, tanshinones are a group of active natural abietane diterpene ingredients isolated from the rhizome of *Salvia miltiorrhiza* Bunge (Danshen) [15,16], a famous traditional Chinese medicine for multiple therapeutic remedies [17]. Previous studies [18,19] indicated that there are more than ten monomers (Appendix A), such as tanshinone I, tanshinone IIA, tanshinone II B, cryptotanshinone, and 1,2-dihydrotanshinquinone, in the rhizome of *S. miltiorrhiza* [15,20]. Several studies indicated that tanshinones have potent biological activities, such as anti-cancer [21], anti-oxidation, cardiovascular pharmacology, antibacterial, and anti-inflammatory [22]. For example, tanshinone IIA was found to inhibit cell proliferation [23], induce cell apoptosis [24], and arrest the cell cycle at the G2/M phase [25,26]. Tanshinone I induced apoptosis in monocytic leukemia U937 THP−1 and SHI 1 cells by means of the activation of caspase-3 and decreasing hTERT mRNA expression [27]. Cryptotanshinone inhibited the expression of cyclin D1, and the phosphorylation of Rb in human rhabdomyosarcoma Rh30 cells induced cell cycle arrest at the G0/G1 phase [28]. In addition, previous studies [21,22,29] also showed that tanshinones could inhibit cell migration [30] through a wide range of signaling pathways, i.e., the HIF-1α-EMT pathway [31], STAT3-CCL2-EMT [32], PI3K/AKT signaling [33], and the PI3K and AMPK-mTOR signaling pathway [34]. However, little is still known about the roles and mechanisms of tanshinones, including dihydrotanshinone I in the regulation of cell adhesion and cell migration [33,35,36,37,38,39].

In view of the significant roles of cell adhesion in cancer progression and metastasis [40,41,42], this work aims to investigate the potent roles and mechanisms of dihydrotanshinone I (15,16-dihydrotanshinone I, DS−1) in enhancing cell adhesion and inhibiting migration in human osteosarcoma U−2 OS cells with a transcriptome-guided identification strategy. After RNA-seq analysis and subsequent biochemical experiments, the results indicated that dihydrotanshinone I induced comprehensive signaling regulation involving CD44 and chemokines.

## 2. Results

### 2.1. Tanshinones Inhibited Cells’ Proliferation

Our previous studies [43] showed that tanshinones could inhibit the proliferation of various tumor cells in vitro including estrogen-receptor-negative breast cancer cells. To verify the effects on proliferation inhibition by five tanshinones (Figure 1) on human osteosarcoma U−2 OS cells, the MTT assay was carried out. As shown in Figure 2A and Table 1, all tanshinones showed potent effects on cell proliferation for 24 and 48 h under the desired dose concentration. Among these tanshinones, dihydrotanshinone I (15,16-dirhydrotanshinone I, DS−1) showed the strongest inhibition effect, and its IC_50_ value for U−2 OS cells for 24 h of treatment was 3.83 ± 0.49 μM, while that for 48 h of treatment was 1.99 ± 0.37 μM, which provided the reference of dosage for subsequent experiments.

It should be pointed out that without treatment of DS−1, common U−2 OS cells as the control grew rapidly, showing a smooth cell surface and clear cell boundaries. However, after being treated with DS−1 at 2.5 and 5.0 µM for 24 and 48 h, the cell surface was no longer smooth; the morphology became irregular. The most important thing was that cell adhesion of DS−1-treated cells on the cell culture plate became significantly stronger than the untreated ones, and cells were tightly attached onto the cell plate, implying the possible roles of DS−1 in cell adhesion and cell migration inhibition [40,41,42].

In addition, as shown in Figure 2B, DS−1 also showed the strongest cytotoxicity against human cervical cancer cells HeLa with an IC_50_ of 15.48 ± 0.98 µM (Table 1). However, DS−1 showed weaker cytotoxicity against normal rat kidney cells NRK−49F (IC_50_ 25.00 ± 1.98 µM, Figure 2C and Table 1). Meanwhile, DS−1 also showed significant cell adhesion on the cell plate, especially against NRK−49F cells, a fibroblastic clone of normal rat kidney cells. Therefore, these phenomenon attracted us to make a deeper investigation.

### 2.2. DS−1 Inhibited the Expression of Transcription Factor FOXO1 in U−2 OS Cells

Cell proliferation involves multiple complex signal regulations. Our previous study [43] indicated that tanshinones could reduce the expression of the FOXO1 transcription factor in human breast cancer Bcap37 cells. FOXO1 is one of the most important transcription factors participating in the cell cycle, cell proliferation, and apoptosis [44,45]. Thus, FOXO1 in U−2 OS cells was detected by immunofluorescence and immunoblotting. After treatment with DS−1, the expression of FOXO1 was reduced (Figure 3A,B).

At the same time, phospho−FOXO1 was also reduced with the concentration of DS−1 and the time of treatment (Figure 3B–F). Meanwhile, when the expression of FOXO1 and phospho-FOXO1 was calculated by Image J, it was found that the phosphorylation of FOXO1 was increased with an increasing concentration of DS−1 (Figure 3G). Increased phosphorylation of FOXO1 indicated that the transcription of U−2 OS cells was depressed by DS−1, further resulting in the inhibition of the growth of U−2 OS cells.

### 2.3. RNA-Seq Gene Expression Profiling in U−2 OS Cells Treated with DS−1

Then, in order to understand the transcriptional landscape alterations during DS−1 treatment, we subjected U−2 OS cells treated with/without 6.25 µM of DS−1 to RNA sequencing gene expression analysis. As shown in Figure 4A,B, the results showed that, in total, 23,517 genes were detected, and among these, 4104 genes were upregulated, while 4643 genes were downregulated. Hierarchical clustering based on differentially expressed RNA transcripts (Figure 4A) revealed clear clustering of DS−1-treated U−2 OS cells from the control cells.

To further explore the roles of these genes, the Kyoto Encyclopedia of Genes and Genomes (KEGG) pathway enrichment analysis and Gene Ontology (GO) term analysis were performed, and the results are illustrated in Figure 4C,D. For KEGG term analysis, it demonstrated that DS−1 had significant effects on focal adhesion, apoptosis, the cell cycle, and the pathway in cancer (Figure 4C).

### 2.4. DS−1 Downregulated Cell Cycle Regulators of U−2OS Cells

As shown in Figure 4C,D, KEGG and GO analyses clearly indicated that DS−1 has a significant impact on the U−2 OS cell cycle. The cell cycle is a key regulator of cell survival and the repair signal network. It has direct effects on multiple cell processes, including cell proliferation, DNA repair, apoptosis, and cell migration [46]. Thus, it was further analyzed by flow cytometry. As shown in Table 2 and Figure 5A–E, after treatment with DS−1 at doses of 0, 2.5, 5.0, and 7.5 µM for 24 h, the distribution of the cell cycle was measured and the proportion of the G0/G1 phase was 54.70 ± 2.21%, 61.07 ± 2.04%, 65.20 ± 4.35%, and 65.65 ± 1.83%, respectively. In addition, the proportion of U−2 OS cells at the G0/G1 phase increased with the increase in the DS−1 concentration (Figure 5A–E), implying possible cell cycle arrest at the G0/G1 phase.

It has been shown that cell cycle transitions are regulated by a comprehensive signaling network. As shown in Figure 5F, the expressions of almost all cell-cycle-related genes were downregulated. Usually, cell cycle transitions are largely dependent on cyclin-dependent kinases (CDKs) and their activating cyclin subunits [47,48]. To analyze the related proteins in the cell cycle, U−2 OS cells were treated with 0, 2.5, and 5.0 µM DS−1 for 24 h, and RT-PCR was carried out. As a result (Figure 5G–J), *CDK2* was downregulated to 0.85 ± 0.08-fold, 0.80 ± 0.16-fold, and 0.66 ± 0.07-fold of the control group; *CDK4* was downregulated to 0.53 ± 0.06-fold, 0.37 ± 0.02-fold, and 0.29 ± 0.05-fold of the control group; *cyclin D1* was downregulated to 0.27 ± 0.06-fold, 0.29 ± 0.06-fold, and 0.20 ± 0.04-fold of the control group; *cyclin E1* was downregulated to 0.52 ± 0.04-fold, 0.48 ± 0.08-fold, and 0.36 ± 0.04-fold of the control group. The downregulation of cyclin D1 and cyclin E1 was further confirmed by immunoblotting (Figure 5L). Downregulation of CDKs, cyclin D1, and cyclin E1 indicated that DS−1 blocked the CDK-related pathway.

The cell cycle is not only regulated by the abundance of cyclin E, but also regulated by p21, which binds the CDK2–cyclin E1 protein complex to make it inactive [49]. Thus, p21 was also analyzed by RT-PCR. U−2 OS cells were treated with 2.5, 5.0, and 7.5 µM DS−1 for 24 h, and *p21* was upregulated to 10.1 ± 3.9-fold, 56.8 ± 8.2-fold and 90.2 ± 6.7-fold of the control group (Figure 5K), further arresting the cell cycle of U−2 OS cells.

These results showed that DS−1 could inhibit the expression of the CDK4/CycD and CDK2/CycE complexes and increase the expression of p21, which promoted the phosphorylation and degradation of CDK4/CycD and inhibited its activity.

### 2.5. DS−1 Inhibited U−2OS Cell Migration

U−2 OS cells have a strong migration ability [2,3,4,14,50]. Primary experiments indicated that DS−1 significantly enhanced the cell adherence capacity on cell plates, implying its possible role in cell adhesion and cell migration inhibition. Thus, a wound scratch assay was carried out to check the effect of DS−1 on U−2 OS cell migration. As shown in Figure 6A,B, after incubation with DS−1 at 2.5 μM for 24 and 48 h, the relative mobility of treated cells compared to control cells was 24.8 ± 9.9% and 18.1 ± 6.6%, respectively. When incubated with DS−1 at 5.0 μM for 24 and 48 h, the relative mobility was 17.5 ± 6.6% and 7.3 ± 7.1% that of control cells, respectively, and when incubated with DS−1 at 7.5 μM for 24 and 48 h, the relative mobility was 6.6 ± 0.8% and 4.3 ± 1.8%, respectively. The results suggest that DS−1 had potent inhibition effects on cell migration, even at a low dose of 2.5 μM.

Cell migration has been found to comprehensively relate to the signals of epithelial–mesenchymal transition (EMT) [51,52]. EMT is a key step toward cancer metastasis. The transcriptome result (Figure 6C) implied that the EMT pathway might be involved in the migration inhibition of U−2 OS cells. FASN is the starting point protein, which promotes the phosphorylation and degradation of E-cadherin, and then reduces intercellular contact [53]. FASN plays an important role in the palmitoylation of Wnt1 and the cytoplasmic stabilization of β-catenin, which regulates HIF-1α and then induces EMT [13]. To confirm the result, FASN mRNA was relatively quantitated by RT-PCR. As expected, *FASN* was downregulated to 0.59 ± 0.14-fold, 0.51 ± 0.07-fold, and 0.26 ± 0.01-fold of the control group with treatment with 2.5, 5.0, and 7.5 µM DS−1, respectively (Figure 6D).

The downregulation of FASN will reduce the expression of snail and slug [54], which are key transcription factors of EMT [55,56]. Inhibition of snail and slug will induce EMT and result in simultaneously suppressing the tumor metastasis of cancer cells. Thus, expressions of snail and slug were checked. After treating with 2.5, 5.0, and 7.5 µM DS−1, the expression of *snail* was downregulated to 1.06 ± 0.06-fold, 0.46 ± 0.05-fold, and 0.39 ± 0.03-fold of the control group, and the expression of *slug* was downregulated to 1.06 ± 0.22-fold, 0.58 ± 0.13-fold, and 0.45 ± 0.07-fold of the control group (Figure 6E,F).

Since EMT is critical to cancer metastasis, the decrease of FASN indicated the hypothesis that DS−1 could inhibit the EMT pathway. Therefore, we checked the marker proteins of the EMT pathway. When the EMT pathway is inhibited, its marker protein, vimentin, will be suppressed, while E-cadherin will be upregulated [57]. To confirm the change in the EMT pathway, U−2 OS cells were treated with DS−1 at doses of 2.5, 5.0, and 7.5 µM for 24 h. As a result, the expression of *vimentin* was downregulated to 0.67 ± 0.11-fold, 0.51 ± 0.11-fold, and 0.12 ± 0.16-fold of the control group, while the expression of *E*-*cadherin* was upregulated to 1.21 ± 0.16-fold, 1.26 ± 0.25-fold, and 1.61 ± 0.16-fold of the control group, (Figure 6G,H), which was further confirmed by Western blot (Figure 6I).

### 2.6. DS−1 Enhanced U−2 OS Cells’ Adhesion

As shown above in Figure 6, after treatment with DS−1, the migration of U−2 OS cells was inhibited significantly. We wondered if there were other pathways affecting cell migration besides EMT signals. During the collection of cells in the former experiments, we found that the time of trypsin digestion of the DS−1 group was significantly longer than the control. As shown in Figure 7A,B, the control group was digested with trypsin within 120 s at 25 °C, while the DS−1 group was digested within 210 s. The results suggested that DS−1 had a potent role in enhancing cell adhesion on the cell culture plate.

Interestingly, in some other cancer cell lines including HeLa cells (Appendix A), the phenomena of cell adhesion against trypsin digestion on cell culture plates were not significant, but in the cells of NRK−49F, a fibroblastic clone of normal rat kidney cells, DS−1 showed stronger cell adhesion on the cell culture plate (Appendix A). These interesting phenomena further attracted us to investigate its special molecular mechanism.

In the transcriptome result, almost all genes relating to the adherens’ junction were upregulated (Figure 7C). Among these upregulated genes, there was a 2.94 ± 0.55-fold upregulation of CD44 in the DS−1 group. It is well known that CD44 is a key receptor in cell adhesion and migration [58,59]. Thus, the expression of CD44 was re-evaluated by immunofluorescence and RT-PCR. As shown in Figure 7D,E, CD44 expression was significantly increased after U−2 OS cells were treated with DS−1 at doses of 0, 2.5, 5.0, and 7.5 µM for 24 h.

Normally, CD44 upregulation increases the transfer of cells and stimulates the invasive ability by interacting with matrix metalloproteinases (MMPs), which localize on the cell surface and mediate the activation of progelatinase A (named membrane-type matrix metalloproteinases (MT-MMPs) [60]. According to the transcriptome result, the expression of different *MT*-*MMPs* is shown in a heatmap (Figure 7G), and all kinds of MT-MMPs were downregulated. Among these MT-MMPs, MT-MMP-1 is most commonly overexpressed in malignant tumor tissues and interacts with CD44 to stimulate the invasive ability. MT-MMP-1 acts as a processing enzyme for 80 kDa CD44, releasing it into the medium as a soluble 70 kD fragment, which will promote cell migration [61]. Thus, CD44 proteins were checked by the Western blot assay. As shown in Figure 7F, after U−2 OS cells were treated with DS−1 at doses of 0, 2.5, 5.0, and 7.5 µM for 24 h, 80 kDa CD44 was significantly increased, but 70 kDa CD44 decreased. Meanwhile, RT-PCR indicated that the expression of *MT*-MMP-*1* was downregulated to 0.92 ± 0.14-fold, 0.72 ± 0.03-fold, and 0.07 ± 0.13-fold after treatment with 2.5, 5.0, and 7.5 µM DS−1, respectively (Figure 7H). These results confirmed that the downregulation of MT-MMP-1 increased 80 kDa CD44 and decreased 70 kDa CD44, resulting in enhanced cell adhesion induced by CD44.

To validate the pivotal role of CD44 in increasing cell adhesion, CD44 in U−2 OS cells was largely knocked down by siRNA (Appendix A). As shown in Figure 8A,B, after being treated with 7.5 μM DS−1 for 24 h, the immunofluorescence of CD44 of the control group was significantly increased, while that of the CD44 of the knockdown group increased a little.

Previous studies [42,59] indicated that, when CD44 was knocked down, the cells became rounder and protrusions became shorter, and cell adhesion to ECM decreased, which meant that the overexpression of CD44 causes cell growth with more and longer protrusions. Actin promotes the formation of protrusions around the cells. After CD44 knockdown and DS−1 treatment, the morphology of actin was observed. The average length of protrusions was measured by Image J [59]. As a result, listed in Figure 8C–E, in the control group, the average length of protrusions was 0.307 ± 0.097 μm; after treatment with DS−1, protrusions were extended to 0.403 ± 0.103 μm. In the CD44 knockdown group, the average length of protrusions was shortened to 0.203 ± 0.056 μm. After treatment with DS−1, the average length of protrusions was only increased to 0.213 ± 0.061 μm. The changes of protrusions’ length were related to the expression of CD44, which was influenced by DS−1 treatment or CD44 knockdown.

After CD44 knockdown by siRNA, another cell digestion experiment was conducted at 25 °C again. In the control group, U−2 OS cells were totally digested with trypsin after about 120 s, while in the DS−1 group, U−2 OS cells were totally digested within 150 s (Figure 8F,G). Compared to the no CD44 knockdown group treated with DS−1, 60 s of digestion time was saved, which indicated that cell adhesion was decreased. From two experiments of digestion, we arrived at the conclusion that DS−1 could increase the adhesion capacity of U−2 OS cells by inducing overexpression of CD44, as well as reducing the expression of MT-MMP-1.

### 2.7. DS−1 Activated CXCL8-PI3K/AKT Signaling Pathway

With the prompt of the transcriptome, CXCL8 is a key gene upregulated by DS−1. Therefore, we evaluated the expression of CXCL8. After treatment with 2.5, 5.0, and 7.5 µM DS−1, *CXCL8* was upregulated to 1.49 ± 0.10-fold, 3.34 ± 0.49-fold, and 3.71 ± 0.53-fold (Figure 9A).

CXCL8 induces rapid and transient phosphorylation of extracellular-signal-related kinases (ERK1/2) and phosphatidylinositide 3-kinase (PI3K)/Akt [62], which is a kind of phosphokinase of FOXO1. AKT and pAKT in U−2 OS cells treated with DS−1 were analyzed by Western blot; no significant change was found in the expression of AKT, but pAKT increased to 1.54 ± 0.10-fold, 2.06 ± 0.08-fold, and 2.10 ± 0.15-fold of the control group with treatment with 2.5, 5.0, and 7.5 µM DS−1. The pAKT/AKT value increased with the concentration of DS−1, which showed that the phosphorylation level was upregulated (Figure 9B,D).

### 2.8. DS−1 Changed IL6-STAT3-P53 Signal

From the transcript result, *IL6* was downregulated. After treatment with 2.5, 5.0, and 7.5 µM DS−1, the expression of *IL6* was downregulated to 0.81 ± 0.09-fold, 0.51 ± 0.07-fold, and 0.42 ± 0.12-fold (Figure 10A).

The JAK/STAT signaling pathway is downstream of IL6, which transfers the signal from many of the cytokines and growth factor receptors to the target genes in the nucleus [32]. Tumor cells contain a large number of active phosphorylated STAT3. STAT3 plays a crucial role in mediating oncogenic transformation and inducing tumor formation. As a cytokine and growth factor downstream protein, STAT3 is phosphorylated by related kinase and then forms homologous or heterodimers, which transfer to the nucleus as an intranuclear transcriptional activator. STAT3 mediates the expression of multiple genes to cell stimuli in tumor cells, thus playing a key role in many cellular processes such as cell growth and apoptosis. In this study, U−2 OS cells were treated with 2.5, 5.0, and 7.5 µM DS−1, and STAT3 was analyzed by Western blot (Figure 10B). The total protein content of STAT3 did not change significantly, but phospho-STAT3 was downregulated to 0.59 ± 0.09-fold, 0.43 ± 0.03-fold, and 0.32 ± 0.03-fold of the control group (Figure 10C). Thus, the ratio of pSTAT3/STAT3 was significantly decreased (Figure 10D).

A recent study indicated that the tumor suppressor gene wildtype p53 is an important regulator in the regulation axis of IL6-STAT3-CD44 [63]. p53 plays a key role in many cellular pathways controlling cell proliferation, cell survival, and genomic integrity. P53 acts as a brake on proliferation when cells experience stress, such as DNA damage, hypoxia, and oncogene activation. Thus, the level of p53 was checked. As shown in Figure 10B, after treatment with DS−1, p53 significantly increased. Increasing p53 subsequently inhibited cell proliferation and regulated the CD44 level, and thus regulated the cell adhesion and migration [63].

## 3. Discussion

Osteosarcoma is the most common primary bone cancer, which occurs primarily in children and adolescents. Despite its chemosensitivity and treatment advances, limited progress has been made in improving the overall survival in patients with osteosarcoma over the past four decades [2,3], due to the tumor cells having a strong ability related to invasion [50], metastasis, rapid proliferation [14], and enhancing the glycolytic pathway, which makes cells resistant to oxidative stress and adaptive to hypoxic conditions [64]. In this work, we found that tanshinones could potently inhibit human osteosarcoma U−2 OS cell proliferation. In particular, DS−1 could enhance cell adhesion on the cell culture plate and inhibit cell migration. This was an interesting phenomenon, implying the special molecular mechanism of DS−1 in U−2 OS cells.

Cell proliferation and cell migration are regulated by a comprehensive network. Transcriptomic analysis indicated that after incubation with DS−1, the gene expressions of U−2 OS cells showed significant changes involving wide signals, such as focal adhesion, the pathway in cancer, thermogenesis, and the cell cycle.

The cell cycle is a series of tightly regulated molecular events controlling DNA replication and mitosis, producing two new daughter cells from a single parent cell. At the beginning of the G1 phase, cyclin D came out, which induced the expression of CDK2, CDK4, and CDK5, then CDK2-cyclin D, CDK4-cyclin D, and CDK5-cyclin D were formed, promoting cells through the G1 restriction point [65]. Meanwhile, cyclin E cumulative synthesis bound with CDK2 to form the CDK2–cyclin E complex, and MCM, the promoter factor of DNA replication, was phosphorylated and active, inducing cells to go to the S phase from the G1 phase [66]. The premise of cell proliferation is cell division. The cell cycle is not only regulated by the abundance of cyclin E, but also regulated by the activity of the cyclin E–CDK2 complex, which is further regulated by p21 and p27. p21 could bind to the CDK2–cyclin E1 complex, which inactivates the CDK2–cyclin E1 complex [49]. In this work, DS−1 downregulated CDK2, CDK4, cyclin D1, and cyclin E1, together with upregulating p21, while DS−1 also possibly arrested the cell cycle at the G0/G1 phase. The heatmap in Figure 3F shows that DS−1 induced the downregulation of most cell cycle regulators. After treatment with DS−1, U−2 OS cells were arrested at the cell cycle phase. This phenomenon revealed that DS−1 inhibited cell division dose-dependently.

Invasion and metastasis are important reasons for the difficulty of curing human osteosarcomas. In this work, the migration inhibition effect of DS−1 on tumor cells was obvious. Epithelial–mesenchymal transition (EMT) is a key step toward cancer metastasis. EMT plays an important role in the migration inhibition of U−2 OS cells. FASN is the starting point protein, which promotes the phosphorylation and degradation of E−cadherin and then reduces intercellular contact [53]. DS−1 decreased the expression of FASN, snail, and slug, thereby inhibiting EMT. The change in the marker molecules E-cadherin and vimentin confirmed that EMT was inhibited by DS−1. In Sung-Ying Huan’s studies, it was found that tanshinone IIA inhibits EMT in bladder cancer cells via modulation of STAT3-CCL2 signaling [32]. Due to the similarity of tanshinone IIA and dihydrotanshinone I (DS−1), tanshinones have the capacity for cell migration inhibition by blocking the EMT pathway and reducing the phosphorylation of STAT3.

Since the migration of U−2 OS cells was significantly inhibited, we posited that another pathway related to migration is regulated by DS−1. Previous studies suggest that MT-MMP-1 cleaves CD44 and promotes cell migration [61,67]. The CD44 receptor interacts with P-glycoprotein and promotes cell migration and invasion in cancer [68]. In this work, the upregulation of CD44 firstly interested us. Furthermore, we found that MT-MMP-1 was downregulated by DS−1. Therefore, it might be concluded that DS−1 also decreased cell migration via the inhibition cleavage of CD44. Except for the adhesion to the matrix, CD44 overexpression could cause more long protrusions, which facilitate adhesion and invasion into the tissue. When CD44 was knocked down, the cells became rounder and the protrusions became shorter, and the adhesion of the cell to ECM decreased, while the overexpression of CD44 caused cell growth with more and longer protrusions [59]. In addition, as an important protein in cancer cells, CD44v6 in combination with HA also promotes the PI3K/AKT signaling pathway and increases apoptosis [69]. Activated STAT3 has been shown to induce matrix metalloproteinase (MMP-2, MMP-9, and MMP-14) expression [70,71,72], which can cleave CD44. These are important factors involved in the degradation of the extracellular matrix necessary for tumor invasion and metastasis [73]. Therefore, DS−1 may be a potent enhancer of cell adhesion to inhibit cell migration.

DS−1 also induced the changes in chemokines, including CXCL8 and IL6, following with the decrease of the phosphorylation of AKT and STAT3. As a result, the cell proliferation and cell cycle were inhibited. In Xiaoqing Wang’s work, the proliferation and migration of ovarian cancer cells were inhibited by dihydrotanshinone I via transcriptional repression of the PIK3CA gene [33]. Although ovarian cancer cells and osteosarcoma cells are different cancer cells, after treatment with the same drug (DS−1), the same inhibition effects on cell migration and cell proliferation were found for the same pathway, which implies that DS−1 possibly targets the same proteins in different cells.

According to these results, we proposed a new and systematic signal pathway in U−2 OS cells induced by DS−1. As illustrated in Figure 11, the major signaling processes included inhibiting cell division, cell proliferation, and migration and enhancing cell adhesion. In general, the results demonstrated that the effects of DS−1 in enhancing U−2 OS cell adhesion and inhibiting proliferation and migration were largely through three major pathways, including the downregulation of cell transcript factors, cell cycle arrest, and cell migration reduction, although it is also involved in cell autophagy, DNA replication, thermogenesis, proteoglycans, endocytosis, and other pathways in DS−1-treated U−2 OS cells, as revealed by the above transcriptome analyses, which require more studies in the future.

As described above, tanshinones are a group of bioactive abietane diterpenes in the rhizome of *Salvia miltiorrhiza* [21,22,29]. They have wide biological activities to inhibit cell growth and migration. For example, Tanshinone IIA could regulate fibroblast proliferation, migration, and post-surgery arthrofibrosis through the autophagy-mediated PI3K and AMPK-mTOR signaling pathway [34]. Dihydrotanshinone I might modulate EMT and subsequently impair the migration and clonogenicity of triple-negative breast cancer cells [37]. Dihydrotanshinone I could inhibit ovarian cancer cell proliferation and migration by transcriptional repression of the PIK3CA gene [33]. Cryptotanshinone might inhibit proliferation and induce apoptosis via mitochondria-derived reactive oxygen species involving FOXO1 in estrogen-receptor-negative breast cancer Bcap37 cells [43]. Tanshinone IIA was found to inhibit EMT in bladder cancer cells via modulation of the STAT3-CCL2 signaling pathway [32]. Thus, STAT3-EMT and PI3K/AKT-FOXO1 might be major regulators of signaling in many cells for tanshinones. Although their chemical structures are very similar, their roles are still somewhat different. In this study, we found that 15,16-dihydrotanshinone I (DS−1) had the most potent capacity to enhance cell adhesion and inhibit cell migration. Besides the STAT3-EMT and PI3K/AKT-FOXO1 signaling pathways, we found that new signals including CD44 and CXCL8 were also involved in DS−1-induced physiological regulation in human osteosarcoma U−2 OS cells.

## 4. Materials and Methods

### 4.1. Materials

Five pure tanshinone compounds, DS−1 to DS−5 (purity, more than 95% by HPLC and ^1^H NMR), were isolated and purified from the extracts of the root of *S. miltiorrhiza* Bunge using the gradient counter-current chromatography method in our lab [18]. Their chemical structures are illustrated in Figure 1. It should be pointed out that 10 mM of each compound in DMSO stored at −20 °C was used as the stock solution and diluted to the desired concentration by the medium before use. For primary cytotoxic MTT screening, the doses were 6.25, 12.5, 25, and 50 µM. Due to these doses being far more than the IC_50_ of DS−1, smaller doses of 2.5, 5.0, and 7.5 µM were used in most of the following experiments, and doses of 3.125, 6.25, and 12.5 µM were used for the experiments regarding FOXO1 in order to collect more surviving cells (at least >15%) for further biochemical experiments. We are very sorry for the inconsistency of some experimental doses because of nontargeted step-and-step investigation of the molecular mechanisms during a long time (more than 3 years). It is indeed a flaw in this work, but does not affect the whole conclusion.

3-(4,5-dimethyl-2-thiazolyl)-2,5-diphenyl-2-H-tetrazolium bromide (MTT), phosphatase inhibitors, phenylmethylsulphonyl fluoride (PMSF), and ammonium persulfate (APS), (TEMED) were purchased from Beyotime Biotechnology (Shanghai, China). Defatted milk powder was purchased Dawen biological Co., Ltd. (Hangzhou, China.). Triton X-100 was purchased from Sangon Biotech Co., Ltd. (Shanghai, China). The BCA protein assay kit was purchased from Thermo Electron Co. (Rockford, USA.) SuperEnhanced chemiluminescence detection reagents and RIPA lysis buffer were purchased from Applygen Technologies Inc. (Beijing, China). Antifade mounting medium with DAPI and RNAeasy™ Animal Long RNA Isolation Kit were purchased from Beyotime Biotechnology (Shanghai, China).

Human osteosarcoma cells U−2 OS, human cervical cells HeLa, and normal rat kidney cells NRK−49F were brought from the cell bank of the Chinese Academy of Sciences (Shanghai, China).

### 4.2. Procedures

#### 4.2.1. Cell Proliferation Assay (MTT)

The effects of tanshinones (DS−1 to DS−5) on the proliferation of U−2 OS cells, HeLa cells, and NRK−49F cells were measured by the classical MTT method with minor modifications as follows. Cells were seeded at 5000 per well in 96-well plates, with each group being repeated in 5 wells, and were incubated for 12 h. DS−1 to DS−5 were added at gradient concentrations, then incubated for 24 and 48 h, respectively. Then, 20 μL MTT solution (solved in PBS, 5 mg/mL) was added to every well, and the cells were further incubated at 37 °C for 4 h until a purple precipitate was visible. The plate was centrifuged at 1000 rpm for 1 min; the supernatant was suctioned, and 150 μL of DMSO was added per well; the plate was shaken on a shaker protected from light for 15 min. We placed the plate in an enzyme marker and measured the absorbance values at a 490 nm wavelength.

#### 4.2.2. Transcriptome Analysis

Sample preparation: U−2 OS cells were seeded in a 10 cm culture dish, and we cultured them to the logarithmic growth phase, suctioned the previous medium, and added new medium with DS−1 at 0 and 6.25 μM; each group was repeated two times. After 24 h of culture, the dish was washed with PBS reagent 3 times. One milliliter of TRIzol reagent was added to every dish for cell lysate and then collected in centrifuge tubes.

Transcriptome sequencing and analysis were carried out by Novogene Bioinformatics Technology Co., Ltd. (Beijing, China).

Raw data in the fastq format were firstly processed through in-house Perl scripts. In this step, clean data were obtained by removing reads containing adapter or containing ploy-N and low-quality reads from raw data. All the downstream analyses were based on the clean data with high quality.

Reference genome and gene model annotation files were downloaded from the genome website directly. The index of the reference genome was built using Hisat2 v2.0.5 (https://github.com/DaehwanKimLab/hisat2, accessed on 4 May 2022), and paired-end clean reads were aligned to the reference genome using Hisat2 v2.0.5. FeatureCounts v1.5.0-p3 (https://github.com/byee4/featureCounts, accessed on 4 May 2022) was used to count the reads numbers mapped to each gene. Then the Fragments PerKilobase of transcript sequence per Millions base (FPKMs) of each gene were calculated based on the length of the gene and reads’ count mapped to this gene.

Differential expression analysis of two groups (each of two biological replicates of control and DS−1 treated) was performed using the DESeq2 R package v1.16.1 (https://github.com/mikelove/DESeq2, accessed on 4 May 2022). DESeq2 provides statistical routines for determining differential expression in digital gene ex-pression data using a model based on the negative binomial distribution. The resulting *p*-values were adjusted using the Benjamini and Hochberg approach for controlling the false discovery rate. Genes with an adjusted *p*-value < 0.05 found by DESeq2 were assigned as differentially expressed.

Gene Ontology (GO) enrichment analysis and KEGG enrichment analysis of differentially expressed genes were implemented by the ClusterProfiler R package v3.4.4 (https://github.com/YuLab-SMU/clusterProfiler, accessed on 4 May 2022).

#### 4.2.3. RT-PCR

Total RNA within cells was extracted by the RNAeasy™ Animal Long RNA Isolation Kit with Spin Column (Shanghai, Beyotime); the concentrations and OD260/OD280 and OD230/OD260 values were measured by NanoDrop. Five micrograms of RNA was used for reverse transcription; each sample was diluted to 16 μL, and 4 μL of synthesized premix solution (5×) was added, then the mixtures were placed in a 42 °C water bath for 10 min, then an 80 °C water bath for 10 min to deactivate reverse transcriptase. After reverse transcription, each sample was diluted to 140 μL. We designed the primer sequence of a gene on the primer3 web version 4.1.0, and the primers are shown in Table 3. PCR amplification was carried out by the ABI700 real-time PCR system, and the amplification reaction was detected with a SYBR probe. After PCR, actin and GAPDH were used as internal references, and the relative standard curve method (2^−ΔΔcT^) determined the relative expression of mRNA.

#### 4.2.4. Scratch Motility (Wound-Healing) Assay

U-2 OS cells were seeded in 6-well plates at a density of 200,000 per well and cultured to 90% abundance after 18 h. The cells were scratched with a 10 μL pipette tip along the ruler; each well was scratched 3 times in parallel, and the line was drawn perpendicular to the scratch on the back for fixed-point photography. We suctioned the previous medium and washed the cells with a PBS reagent. Then, we added 2% FBS medium with DS−1 and cultured the cells at 37 °C for different times. After 0, 24, and 48 h, the plate was taken out, and the scratches were observed and photographed by the inverted microscope, while the distance of scratches at different times was measured.

Cell mobility = (pre-migration distance – post-migration distance)/pre-migration distance × 100%. The cell migration rate of blank control was set as 100%, and the relative cell migration rate = cell migration rate/cell migration rate of control group × 100%

#### 4.2.5. Immunofluorescence

Coverslip preparation: we cultured U−2 OS cells on sterile glass coverslips in 24-well plates and then treated them with DS−1 at different concentrations.

Fixation and staining: We washed the cells with PBS 3 times and fixed the cells with 200 μL 4% paraformaldehyde/PBS solution at 4 °C for 15 min. We then rinsed 3 times with PBS and permeabilized with 200 μL 0.2% TritonX-100 at room temperature for 30 min. Then, we blocked with 200 μL blocking agent at 37 °C for 1 h and diluted the primary antibody in dilution buffer as recommended in the specific products’ datasheet. We overlaid 150 μL of diluted antibody and kept slips covered overnight at 4 °C or for 1 h at 37 °C. We rinsed 3 times with PBS, 5 min for each wash, diluted fluorescent secondary antibody in dilution buffer, and incubated for 1 h at room temperature. Then, we rinsed 3 times with PBS again, 5 min for each wash. We took out the slips and laid them on the slides, dropping 3 μL of antifade mounting medium with DAPI onto them. Then, we covered the slips with cover glass. We then observed and photographed the samples with two-photon laser confocal microscopy (Lsm710nlo).

#### 4.2.6. Western Blot

Sample preparation: U−2 OS cells seeded in 10 cm Petri dishes were collected and washed with ice-cold PBS and RIPA lysis buffer with PMAF; phosphatase inhibitor was added, and the tubes were placed on ice for 30 min and vortexed for 30 s at intervals of 10 min. The samples were centrifuged at 12,000 g at 4 °C for 15 min, and the supernatants were collected in new centrifuge tubes. The protein samples were placed on ice, and the concentration of each protein sample was measured by the BCA kit (Thermo). The protein concentration was adjusted to the same by milli Q ultrapure water.

Loading and running the gel: Equal amounts of protein were loaded into the wells of the 10% SDS-PAGE gel, along with molecular weight markers, and 15 μg of total protein from the cell lysate was loaded. We then ran the gel for 45 min at 200 V.

Transferring the protein: We prepared the transfer stack as follows: negative plate, sponge, filter paper, gel, PVDF membrane, filter paper, sponge, and positive plate. The transfer was for 90 min at 200 mA.

Antibody staining: We blocked the membrane for 1 h at room temperature using 5% blocking solution, then incubated the membrane with appropriate dilutions of the primary antibody in the blocking solution overnight at 4 °C. We then washed the membrane with TBST reagent 3 times, 5 min each. Then, we incubated the membrane with the recommended dilution of labeled secondary antibody in blocking solution at room temperature for 1 h and washed the membrane with TBST reagent 3 times, 5 min each. For signal development, super-enhanced chemiluminescence detection reagents were added to the surface of the membrane for 1 min, then excess reagent was removed, and the membrane was covered in transparent plastic wrap. Images were acquired using darkroom development techniques for chemiluminescence and analyzed by Image J.

#### 4.2.7. Transient Transfection to Knock Down CD44 in U−2 OS Cell

Small interfering RNA (siRNA) oligonucleotide was synthesized by Sunya Biotechnology Co., Ltd (Hangzhou, China). The siRNA sequences were as follows: sense (5′–3′), UAUUCCACGUGGAGAAAAAtt; antisense (5′–3′), UUUUUCUCCACGUGGAAUAca [74]. A total of 5 × 10^5^ cells were plated in 2 mL of growth medium in a 6-well plate and cultured for 12 h until 70–90% confluence was obtained. To transfect each well, 100 pmol siRNA was diluted in 200 µL of opti-MEM without serum, and 5 µL of HighGene Transfection reagent (RM09014, Abclonal) was also added. Then, the solution was mixed by vortexing. SiRNA and HighGene Transfection reagent complexes were added to each well and shaken gently. Half the amount of the used medium was displaced by fresh medium after 6 h of cell transfection in order to reduce the toxicity of the reagents to the cells. After 24 h, the transfected cells were treated with DS−1 or harvested for further use.

#### 4.2.8. Antibodies

Antibodies against FOXO1, phospho-FOXO1 (Ser256), and AKT were purchased from Abcam Biotechnology (Cambridge, USA). Antibodies against CD44, Phospho-AKT (S473), Bcl-2, p53, phospho-p53 (S20), p21, cyclin D1, and cyclin E1 were purchased from Abclonal Biotechnology Co., Ltd. (Wuhan, China). Antibodies against E-cadherin, vimentin, actin, and GAPDH were purchased from Beyotime Biotechnology. Peroxidase-conjugated goat anti-rabbit and anti-mouse IgG (H + L), FITC-conjugated goat anti-rabbit IgG (H + L), and Cy3-conjugated goat anti-rabbit IgG (H + L) were purchased from Dawen biological Co., Ltd. (Hangzhou, China).

#### 4.2.9. Statistical Analysis

Statistical analysis was performed using Graphpad Prism 5.0 (GraphPad Software, Chicago, US). Results are expressed as the mean ± standard deviation (SD) from at least three separate experiments. Student’s *t*-test was used to evaluate the significance between the two groups, while multiple-group comparisons were evaluated by one-way ANOVA followed by Bonferroni’s test. A value of *p* < 0.05 was considered statistically significant, and asterisks show significant differences between groups: * *p* < 0.05, ** *p* < 0.01, *** *p* < 0.005.

## 5. Conclusions

In this study, DS−1 was found to possess potent effects in enhancing cell adhesion and inhibiting cell proliferation and migration on U−2 OS cells. The transcriptome results indicated that DS−1 could regulate a comprehensive signal with multiple biological processes. Further biochemical experiments confirmed that DS−1 could arrest the cell cycle, inhibit cell proliferation and migration, and enhance cell adhesion, mainly through the hyaluronan CD44-mediated CXCL8-PI3K/AKT-FOXO1, IL6-STAT3-P53, and EMT signaling pathways. Therefore, all the results indicated that dihydrotanshinone I (DS−1) could be an interesting molecule for enhancing cell adhesion and inhibiting cell proliferation and migration of U−2 OS cells. To the best of our knowledge, this is the first report that DS−1 can induce CD44-mediated U−2 OS cell adhesion and proliferation inhibition, although more work is required to explore the therapeutic roles using multiple cells and animal models in vitro and in vivo. This is very useful for future osteosarcoma research and drug development.

## Figures and Tables

**Figure 1 molecules-27-03714-f001:**
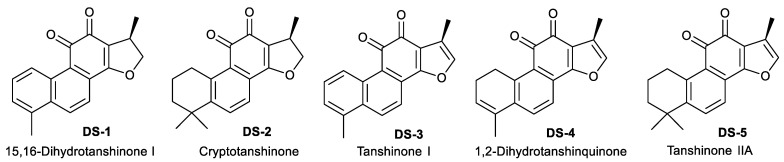
The chemical structures of 5 tanshinones isolated from the rhizome of *Salvia miltiorrhiza* Bunge.

**Figure 2 molecules-27-03714-f002:**
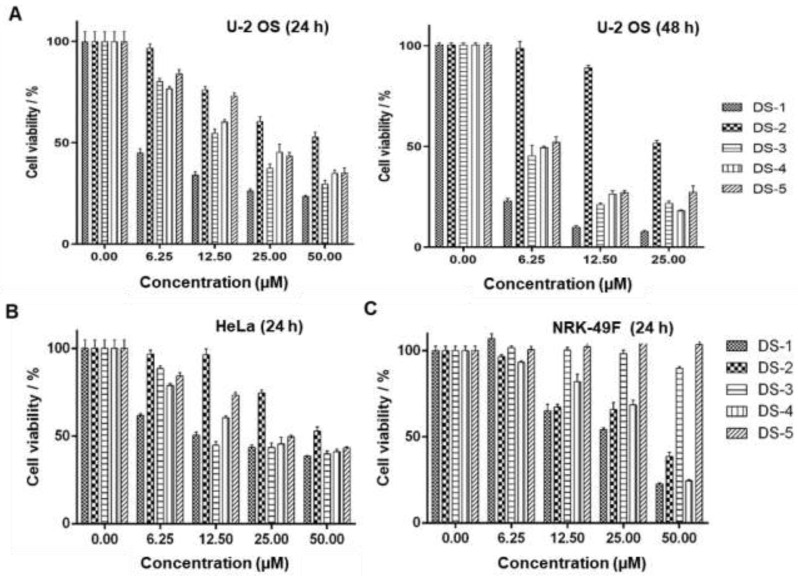
Danshinones inhibited cell proliferation. Cell viability of tanshinone-treated (**A**) U−2 OS cells, (**B**) human cervical carcinoma HeLa cells, and (**C**) normal rat kidney cells NRK−49F. Cancer cells were seeded in a 96-well plate and cultured in the desired tanshinone drugs at different concentrations for 24 and 48 h, and the survival rate was detected by an MTT essay. Data are the mean ± S.D. from three independent experiments with similar results.

**Figure 3 molecules-27-03714-f003:**
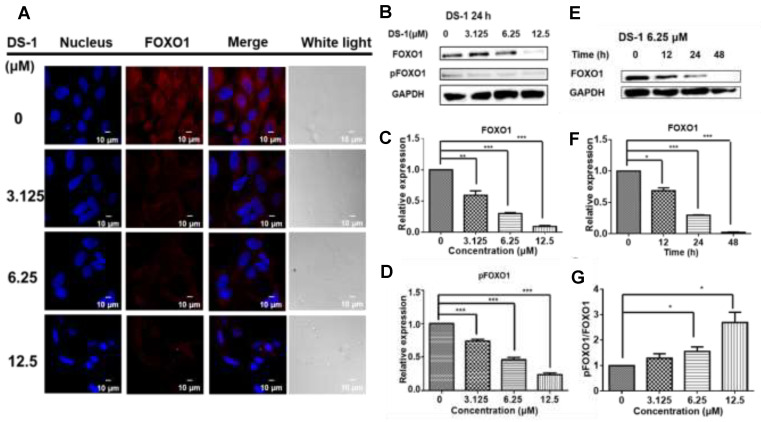
DS−1 inhibited the expression of transcription factor FOXO1 in U−2 OS cells. (**A**) U−2 OS cells were treated with DS−1 at 0, 3.125, 6.25, and 12.5 μM, and the changes in FOXO1 were analyzed by immunofluorescent staining, followed by confocal microscopy imaging. (**B**) U−2OS cells were treated with DS−1 at 0, 3.125, 6.25, and 12.5 μM, and FOXO1 and pFOXO1 were analyzed by Western blot; the changes were presented in histograms (**C**,**D**). (**E**) U−2 OS cells were treated with DS−1 at 6.25 μM for 0, 12, 24, and 48 h, then FOXO1 was analyzed by Western blot, and the changes are presented in the histogram (**F**). (**G**) The pFOXO1/FOXO1 value is shown in the histogram. Values are the average of three independent experiments, * *p* < 0.05, ** *p* < 0.01, *** *p* < 0.005.

**Figure 4 molecules-27-03714-f004:**
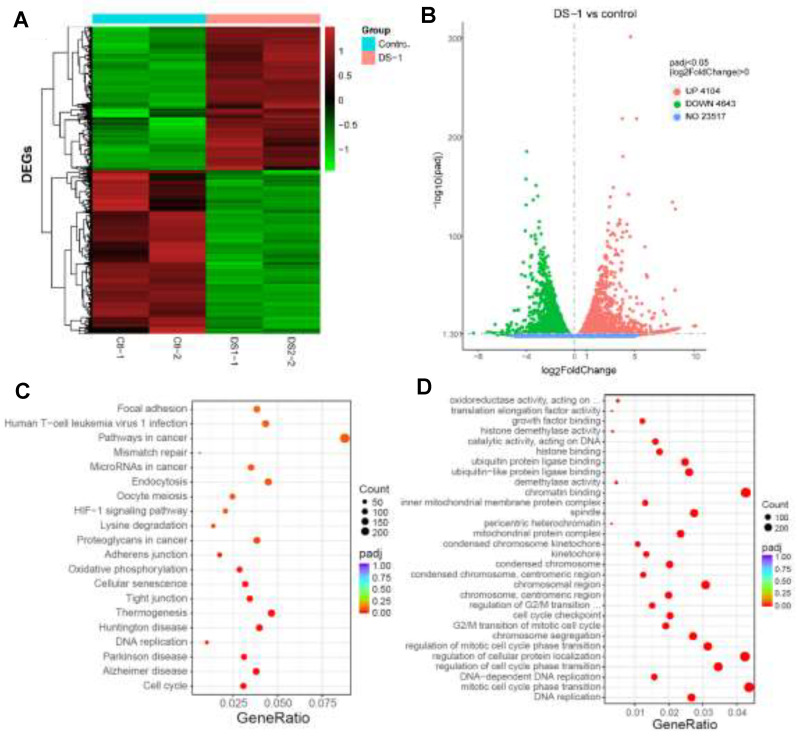
RNA-seq gene expression analyses of U−2 OS cells with/without DS−1 (6.25 µM). (**A**) Heatmap of detected genes differentially expressed in the control group and DS−1 group. (**B**) Volcano map of detected genes differentially expressed in the control group and DS−1 group. (**C**) Kyoto Encyclopedia of Genes and Genomes (KEGG) analysis of differentially expressed genes. (**D**) Gene Ontology (GO) term analysis of differentially expressed genes is shown in the scatter plots.

**Figure 5 molecules-27-03714-f005:**
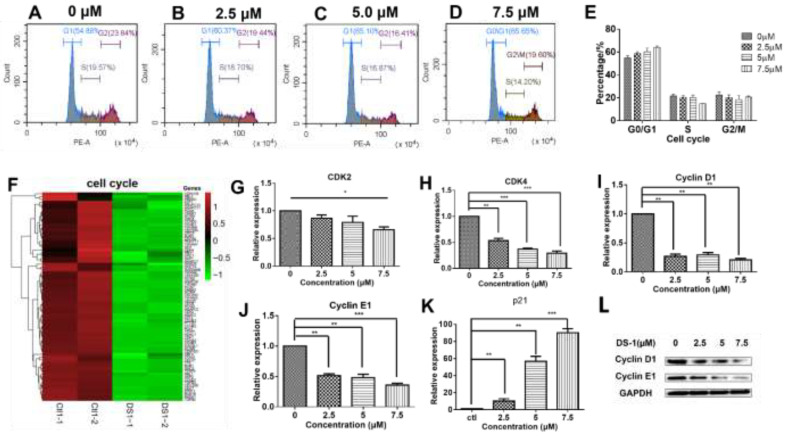
DS−1 regulated the cell cycle of U−2 OS cells. (**A**–**D**) Cell cycle of U−2 OS cells was retained in the G0/G1 phase. U−2 OS cells were treated with DS−1 at 0, 2.5, 5.0, and 7.5 µM for 24 h, and the cell cycle ratio was measured by flow cytometry; the cell cycle distribution is shown in Table 2. (**E**) Histogram of cell cycle distribution after U−2 OS cells were treated with DS−1. (**F**) Heatmap of the expression of genes relating to cell cycle in the transcriptome results. (**G**–**K**) In U−2 OS cells treated with DS−1 at concentrations of 0, 2.5, 5.0, and 7.5 μM, RNA was extracted and analyzed by RT-PCR, and the changes in *CDK*2, *cyclin D*1, *cyclin E*1, and *p*21 mRNA are presented in the histogram. (**L**) Cyclin D1 and cyclin E1 protein relative expression was analyzed by Western blot. Values are the average of three independent experiments, * *p* < 0.05, ** *p* < 0.01, *** *p* < 0.005.

**Figure 6 molecules-27-03714-f006:**
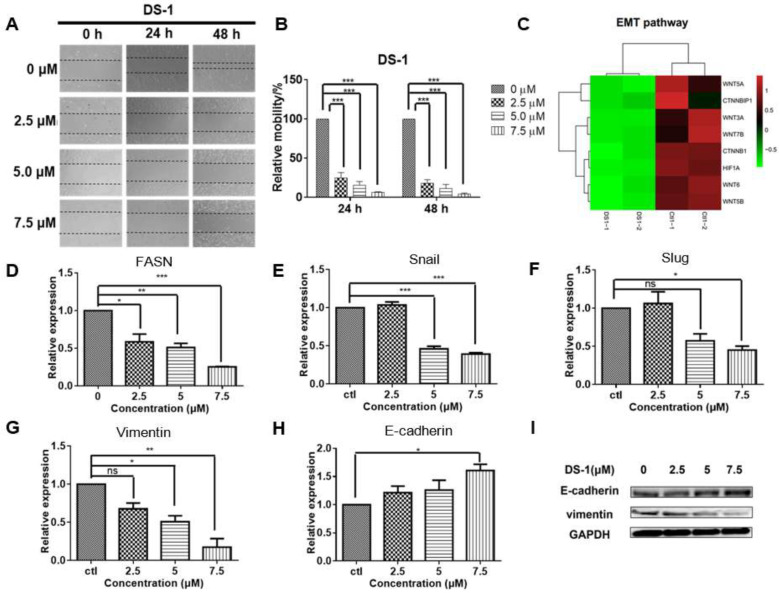
DS−1 inhibited cell migration of U−2 OS cells. (**A**,**B**) U−2 OS cells were cultured to the logarithmic growth phase; the cell monolayers were scratched by pipet tips, and DS−1 at 0, 2.5, 5, and 7.5 µM was added to the plate for 24 and 48 h, respectively. Cell migration was recorded at the same location of the scratch. Data of relative mobility are shown as the mean ± S.D. from three independent experiments. *** *p* < 0.005. (**C**) The expressions of several genes regulating EMT signals from the transcriptome result are shown in the heatmap. (**D**–**H**) U−2 OS cells were treated with DS−1 at 2.5, 5.0, and 7.5 µM for 24 h, and the relative expression of FASN, snail, slug, vimentin, and E-cadherin mRNA are shown in histograms. Values are the average of three independent experiments; * *p* < 0.05, ** *p* < 0.01, *** *p* < 0.005. (**I**) EMT marker molecules E-cadherin and vimentin were analyzed by Western blot.

**Figure 7 molecules-27-03714-f007:**
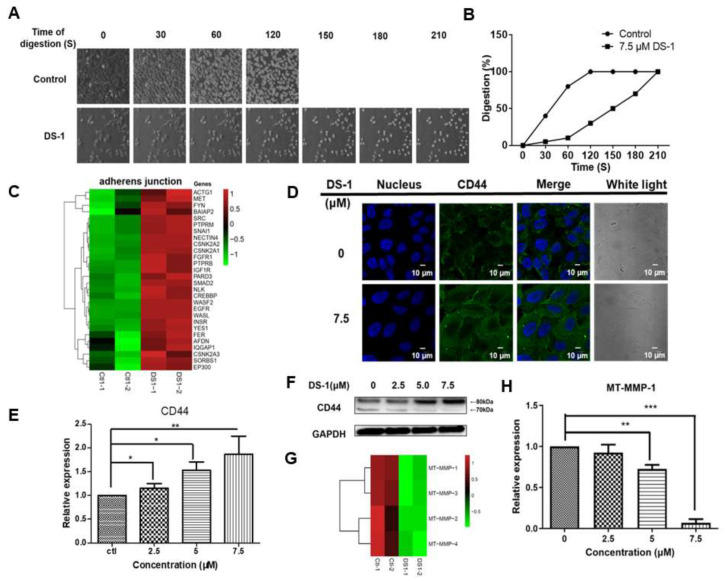
DS−1 enhanced U−2 OS cells’ adhesion. (**A**) The morphological changes of U−2 OS cells treated with/without DS−1. Cells were treated with 0 and 7.5 μM DS−1 for 24 h, washed with PBS, and digested with trypsin at 25 °C under a microscope, and photos were taken every 30 s. The percentages of digested cells are recorded in (**B**). (**C**) Heatmap of genes related to the adherens’ junction in the control group and DS−1 group. (**D**) U−2 OS cells were treated with 0 and 7.5 μM DS−1 for 24 h, and the expression of CD44 was observed by immunofluorescence. (**E**,**F**) CD44 and GAPDH were analyzed by Western blot and calculated by Image J. (**G**) The relative expression of 4 kinds of *MT*-*MMPs* was analyzed by the transcriptome, as shown in the heatmap. (**H**) After being treated with DS−1, the relative expression of *M*T-MMP-1 was verified by RT-PCR. Values are the average of three independent experiments, * *p* < 0.05, ** *p* < 0.01, *** *p* < 0.005.

**Figure 8 molecules-27-03714-f008:**
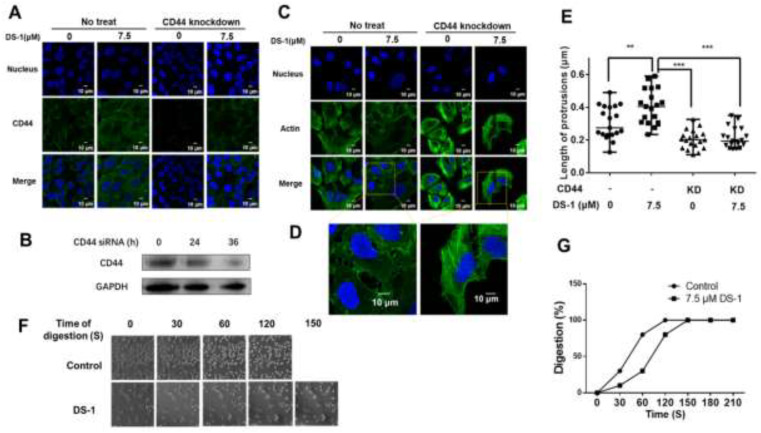
CD44 knockdown reduced cell adhesion by shortening cell protrusions. (**A**) U−2 OS cells transfected with CD44 siRNA and no siRNA were seeded into a 24-well plate treated with 7.5 μM DS−1 for 24 h, and the expression of CD44 was detected by immunofluorescence. (**B**) CD44 expression of U−2 OS cells was detected by Western blotting after being treated with siRNA for 0, 24, and 36 h. (**C**) U−2 OS cells were first transfected with/without CD44 siRNA and then treated with DS−1 for 24 h, and its actin morphologies were detected. The representative magnified actin morphologies are listed in (**D**). (**E**) The lengths of protrusions in (**C**) were analyzed by Image J and are shown in a scatterplot graph. The lengths of 20 protrusions from actin in (**C**) were calculated. Each value is the average of 20 lengths of protrusions, ** *p* < 0.01, *** *p* < 0.005. (**F**) U−2 OS cells were transfected with CD44 siRNA, then treated with 0 and 7.5 μM DS−1 for 24 h and digested with trypsin at 25 °C; the digestion time was recorded by a microscope, and photos were taken every 30 s. The percentage of digested cells is recorded in (**G**).

**Figure 9 molecules-27-03714-f009:**
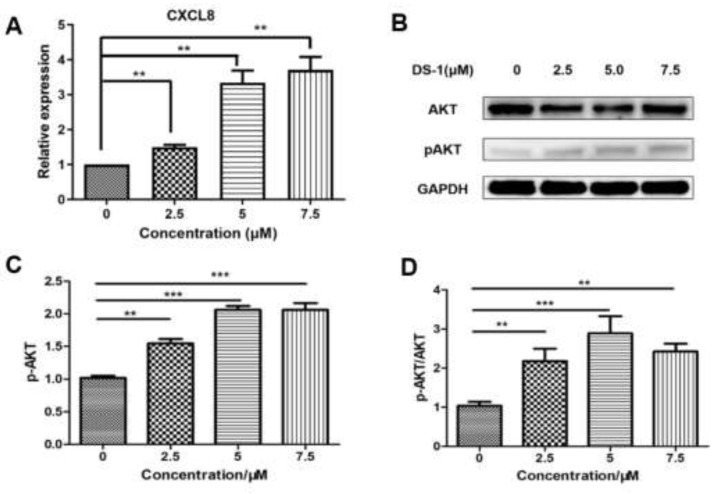
DS−1 induced upregulation of AKT phosphorylation in U−2 OS cells. (**A**) U−2OS cells were treated with DS−1 at 0, 2.5, 5, and 7.5 µM for 24 h, and the change in CXCL8 mRNA was analyzed by RT-PCR and is presented in the histogram. (**B**) U−2 OS cells were treated with DS−1 at 0, 2.5, 5, and 7.5 µM for 24 h, and proteins AKT and p-AKT were analyzed by Western blot. (**C**) The result of the Western blot was calculated by Image J and shown in the histogram. (**D**) The p-AKT/AKT value is shown in the histogram. Values are the average of three independent experiments, ** *p* < 0.01, *** *p* < 0.005.

**Figure 10 molecules-27-03714-f010:**
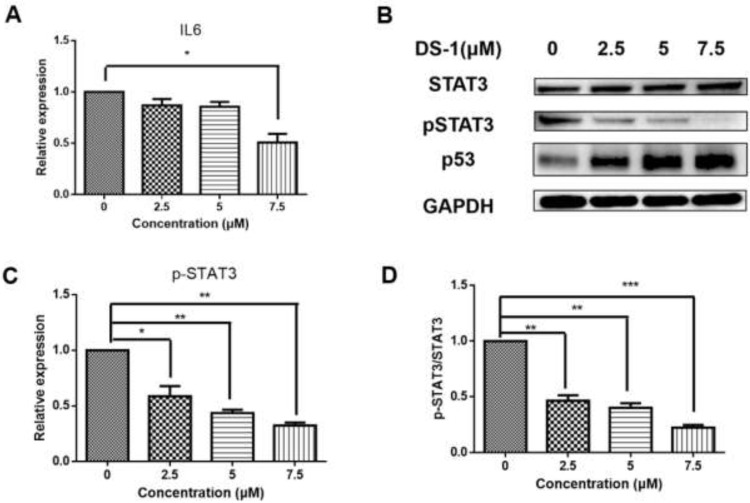
DS−1 inhibited the phosphorylation of STAT3. (**A**) U−2OS cells were treated with DS−1 at 0, 2.5, 5.0, and 7.5 µM for 24 h, and the change in *IL-6* mRNA was analyzed by RT-PCR and presented in the histogram. (**B**) U−2 OS cells were treated with DS−1 at 0, 2.5, 5.0, and 7.5 µM for 24 h, and STAT3, pSTAT3, and p53 were analyzed by Western blot. (**C**) The expression of pSTAT3 is shown in the histogram. (**D**) The pSTAT3/STAT3 value is shown in the histogram. Values are the average of three independent experiments, * *p* < 0.05, ** *p* < 0.01, *** *p* < 0.005.

**Figure 11 molecules-27-03714-f011:**
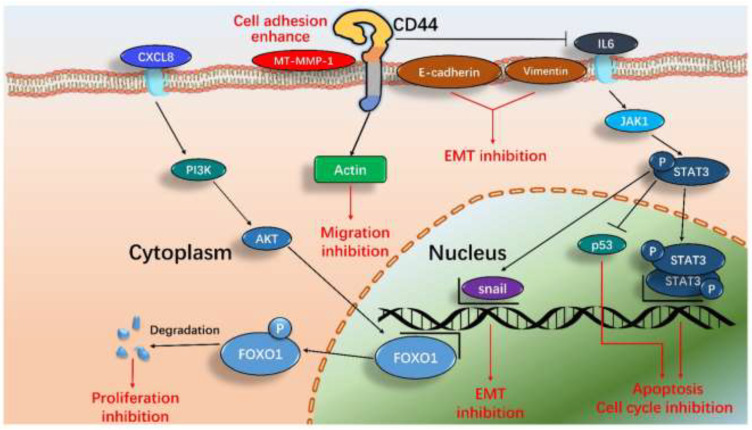
The proposed pathways of major signals induced by DS−1 in U−2 OS cells include inhibiting the EMT pathway, enhancing cell adhesion by CD44 actin protrusions, inhibiting the cell cycle, and inducing cell apoptosis by means of the IL-6-STAT3 signaling pathway and CXCL8-PI3K/AKT-FOXO1 signaling pathway.

**Table 1 molecules-27-03714-t001:** The IC50 values of tanshinones to U−2 OS cells, Hela cells, and NRK−49F cells.

IC_50_ (µM)	DS−1	DS−2	DS−3	DS−4	DS−5
U−2 OS cells (24 h)	3.83 ± 0.49	47.84 ± 4.96	17.75 ± 1.03	22.06 ± 1.31	25.40 ± 1.43
(48 h)	1.99 ± 0.37	25.60 ± 0.81	4.54 ± 1.00	5.89 ± 0.40	5.91 ± 1.08
HeLa cells (24 h)	15.48 ± 0.98	52.07 ± 3.20	20.92 ± 2.81	25.03 ± 2.03	32.09 ± 1.94
NRK−49F cells (24 h)	25.00 ± 1.98	35.39 ± 3.25	107.60 ± 28.85	31.84 ± 1.59	/ ^a^

^a^ no significant cytotoxicity.

**Table 2 molecules-27-03714-t002:** Cell cycle distribution of U−2 OS cells treated with/without DS−1.

DS−1 (µM)	0	2.5	5.0	7.5
G0/G1	54.70 ± 2.21	61.07 ± 2.04	65.20 ± 4.35	65.65 ± 1.83
S	24.00 ± 2.15	20.10 ± 2.07	19.55 ± 3.08	14.20 ± 0.40
G2/M	20.40 ± 3.95	18.65 ± 3.50	14.92 ± 4.67	19.60 ± 1.40

**Table 3 molecules-27-03714-t003:** The design of primers.

Gene	Upstream Primers (5′ to 3′)	Downstream Primers (5′ to 3′)
*GAPDH*	GGAGCGAGATCCCTCCAAAAT	GGCTGTTGTCATCATTCTCATGG
*Actin*	CCGTCTTCCCCTCCATCGT	ATCGTCCCAGTTGGTTACAATGC
*MT-MMP-1*	GGTGGAGGTTGTAGGTGTGA	CAACCCTCAGAGAGCAAAGC
*CDK2*	CCAAGTGAGACTGAGGGTGT	CCAAGTGAGACTGAGGGTGT
*CDK4*	CTGCAGGCTCATACCATCCT	ACTCTTGAGGGCCACAAAGT
*Cyclin D1*	CTCCTTTCTCCACCCACCTC	TCCTCTGCTGGACACCCC
*Cyclin E1*	CAGCGGTTGTAATGTGACCC	AAAGCTCTTCCCCACCCAAT
*CXCL8*	TGTCCTATTGAGAACCACGGT	GCAAGCTAAGACTCTCCAGC
*p21*	CCCAAGCTCTACCTTCCCAC	CTGAGAGTCTCCAGGTCCAC
*IL6*	TCCACTGGAATTTGCTTGCC	AGTGCCTCTTTGCTGCTTTC
*FASN*	TGTGGTGTGTGGGTTGGTAT	GGACGAAATGGGGATAGCCT
*Snail*	CTACCTGTTTGCACACTCGG	ATTTGGTCTTGGCAAAAGCC
*Slug*	TTCTACGTTCTCTGGGCTGG	ACCCAGGCTCACATATTCCT
*E-cadherin*	AACGCATTGCCACATACACTC	GACCTCCATCACAGAGGTTCC
*Vimentin*	TCAATGTTAAGATGGCCCTTG	TGAGTGGGTATCAACCAGAGG

## Data Availability

The datasets generated during the current study are available from the corresponding author upon reasonable request.

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
