# Peer review of "Dihydrotanshinone I Enhances Cell Adhesion and Inhibits Cell Migration in Osteosarcoma U−2 OS Cells through CD44 and Chemokine Signaling"

_molecules, 2022, doi:10.3390/molecules27123714_

Round 1

Reviewer 1 Report

In this manuscript, the authors investigated the biological effects of dihydrotanshinone I, a member of the group of tanshinones used in traditional Chinese medicine, using human osteosarcoma U-2 OS cells as an in vitro cancer cell model.

In particular, the results of the authors that dihydrotanshinone I enhances cell adhesion and inhibits cell migration are of potential interest.  The involvement of different signaling pathways including the hyaluronan CD44-mediated CXCL8–PI3K/AKT–FOXO1-, IL6–STAT3–P53- and EMT pathways is shown.

The results presented by the authors are new and well documented.  They are also of interest to the readers of “Molecules”. However, this paper needs major revision before it can be published in the journal.

My comments are as follows:

My biggest objection is that this manuscript is partially written in poor English. Therefore, this manuscript, which certainly provides a scientifically valuable contribution, has to be extensively rewritten.  The authors should ask an English-speaking colleague for correction. There are also many typos, e.g. “tanshione” instead of “tanshinone” in the Abstract.

The title of the manuscript is misleading and should be correct, e.g. “Dihydrotanshinone I-induced cell adhesion and inhibition of cell migration of osteosarcoma U-2 OS cells involves CD44 and chemokine signaling” or better.

Unfortunately, there are some inconsistencies in the concentrations used by the authors in the dose-response experiments. The author are aware of this and also mention it in the Materials and Methods section of the paper. Perhaps they have more recent results to partially address this weakness.

Figure 5D-H: The level of significance is shown for the concentration of 7.5 µM only. I assume that, e.g., the values for 7.5 µM in Figure 5E,F and G are also significant. This should be specified.

The same applies to Figure 9C – 2.5 µM.

Figure 10: The figure should be corrected: “E-cadherin” instead of “E-cardherin”, and “Vimentin” instead of “Vitmentin”.

There are various papers on partially similar biologically effects of other tanshinones, which are also mentioned by the authors in the Introduction section, e.g. on the effects of tanshinones  on the STAT3-EMT and PI3K/AKT signaling pathways. The authors should also compare and discuss their findings with this previous work in the Discussion section.

In the (sub)headings, lower and upper case letters are used differently, e.g. 4.2.7 Transient Transfection to Knock Down CD44 in U-2 OS cell and 4.2.9 Statistical Analysis, compared  to others.

The references are not consistently given, e.g. sometimes the full names of the journals and sometimes abbreviations are used; in some titles all words are capitalized at the beginning.

Author Response

Comments and Suggestions for Authors

My biggest objection is that this manuscript is partially written in poor English. Therefore, this manuscript, which certainly provides a scientifically valuable contribution, has to be extensively rewritten.  The authors should ask an English-speaking colleague for correction. There are also many typos, e.g. “tanshione” instead of “tanshinone” in the Abstract.

Response: Thank you very much for your kind recommendation and critical suggestions. In deed, there are still many errors including typos and languages although our primary manuscript was edited by MDPI English pre-edit service. We try our best to improve the manuscripts and many errors have been revised. We hope that current revisions will meet the requirements.

The title of the manuscript is misleading and should be correct, e.g. “Dihydrotanshinone I-induced cell adhesion and inhibition of cell migration of osteosarcoma U-2 OS cells involves CD44 and chemokine signaling” or better.

Response: Thank you very much for your valuable comments and critical suggestions. Indeed, previous title is misleading. Aaccording to your suggestion and other references, the title was modified to “Dihydrotanshinone I enhances cell adhesion and inhibits cell migration in osteosarcoma U-2 OS cell through CD44 and chemokine signaling”.

Unfortunately, there are some inconsistencies in the concentrations used by the authors in the dose-response experiments. The authors are aware of this and also mention it in the Materials and Methods section of the paper. Perhaps they have more recent results to partially address this weakness.

Response: Thank you very much for your kind comments and critical suggestions. Indeed, there are several experiments using different concentrations. In the early preliminary cytotoxicity screening, we used the 50 µM as maximum dose (which is the commonly used dose for cancer screening of many natural products) and then diluted to its half dose in turn (25 µM, 12.5 µM, 6.25 µM). After we obtained the IC50 values, we used 2.5 µM, 5 µM and 7.5 µM as later experimental doses. The major reason is that high doses of DS-1 (50 µM, 25 µM, 12.5) made large number of cells death while the concentrations of working solution of DS-1 (2.5 µM, 5 µM and 7.5 µM) are not only near to its IC50 value (3.83 µM), but also easier to dilute from 10 mM DMSO stock solution than previous concentrations such as 12.5 µM and 6.25 µM.

We are very sorry for the inconsistency of some experimental doses because of nontargeted step-and-step invesitgatonn on the molecular mechanisms during a long time (more than 3 years). If united all results, more than half of biochemical experiments should be to redo. It needs much time. In addition, the first author has graduated from Zhejiang University. We hope that that reviewer can agree it. It is indeed a flaw for this work but does not affect the whole conclusion.

Figure 5D-H: The level of significance is shown for the concentration of 7.5 µM only. I assume that, e.g., the values for 7.5 µM in Figure 5E, F and G are also significant. This should be specified.

The same applies to Figure 9C – 2.5 µM.

Response: Thank you very much for careful comments and valuable comments. As you pointed, We checked the data and found that several level of significances were not marked in these figures. We corrected these figures.

Figure 10: The figure should be corrected: “E-cadherin” instead of “E-cardherin”, and “Vimentin” instead of “Vitmentin”.

Response: Thank you very much for your carful corrections. The figures are revised.

There are various papers on partially similar biologically effects of other tanshinones, which are also mentioned by the authors in the Introduction section, e.g. on the effects of tanshinones on the STAT3-EMT and PI3K/AKT signaling pathways. The authors should also compare and discuss their findings with this previous work in the Discussion section.

Response: Thank you very much for your critical suggestions. Indeed, there are several papers on the similar biological effects of other tanshinones, especiallt STAT3-EMT and PI3K/AKT signaling pathway. We add several discussions in the text.

In the (sub)headings, lower and upper case letters are used differently, e.g. 4.2.7 Transient Transfection to Knock Down CD44 in U-2 OS cell and 4.2.9 Statistical Analysis, compared  to others.

Response: Thank you very much for your careful comments. The text has been revised with the first letter of all sub-headings capitalized only.

The references are not consistently given, e.g. sometimes the full names of the journals and sometimes abbreviations are used; in some titles all words are capitalized at the beginning.

Response: Thank you very much for your kind comments and critical suggestions. All references are adjusted to a consistent style, the last name are abbreviated and the first name are full spelled, the first letter of titles are capitalized, and all journals are in the full names.

General Response: Thank editors and reviewers very much for critical reviews and constructive suggestions as well as detailed corrections. We have tried our best to revise our manuscript carefully according to the reviewer’s and editor’s comments. We sincerely hope that the revised manuscript will meet with approval. Once again, thank you very much for your comments and suggestions. If it is still not satisfactory, please contact us ([email protected], 86-13968131541). We will revise it until it is OK.

Reviewer 2 Report

This manuscript needs to include a clear description not only of the folds seen but provide the reader with a clear depiction of what is seen in each figure. Often and nearly all the time the authors state that ## was ##% or ##-fold reduced in Figure #. But they do not describe what is shown in Figure # well enough so that the reader (one who did not do the experiment) can tell what it means. 

MAJOR REVISIONS are required to that all images and plots in the figures can be connected in a logical fashion with the conclusions in the text.

This reviewer has tracked edits in the PDF manuscript to better help the authors submit a revision.

Author Response

Comments and Suggestions for Authors

This manuscript needs to include a clear description not only of the folds seen but provide the reader with a clear depiction of what is seen in each figure. Often and nearly all the time the authors state that ## was ##% or ##-fold reduced in Figure #. But they do not describe what is shown in Figure # well enough so that the reader (one who did not do the experiment) can tell what it means. 

MAJOR REVISIONS are required to that all images and plots in the figures can be connected in a logical fashion with the conclusions in the text.

 Response: Thank you very much for your kind recommendation and critical suggestions. They are very useful for improving our manuscript. All comments were responded and the whole manuscript was revised.  

We need a figure or SI figure to show the structures of ALL compounds in this class and mentioned in the manuscript. Readers should not have to look them up.

 Response: Thank you very much for your critical suggestions. The structures of all tanshinone monomers are displayed in supplementary Figure A1 for readers’ information. We also listed the investigated 5 tanshinones in the revised figure 1.

Two significant figures are not analytically possible, please change these numbers to 3.8±#.# where #.# denotes the standard deviation. DO THIS EVERYWHERE YOU DESCRIBE A FOLD CHANGE

 Response: Thank you very much for your kind suggestions. The value of whole text are revised to the format of “3.8±#.#”, with standard deviation.

Size bars are impossible to read. (Figure 6 and Figure 7)

Please increase the size of the fonts in panels C-G so they are large enough to read.  10-11 arial is idea and ideally they should match the size in B-G as in A. (Actually in most of figures)

Response: Thank you very much for your critical comments. The size bars were adjusted to visible size, and the the size of the fonts in figures are enlarged in order to be seen clearly.

Need some indication as to what this reflects. (Figure 3A)

Need clear labeling of y-axis and its meaning. (Figure 5)

Response: Thank you very much for your key comments and useful suggestions.

The results of Figure 3A show that 4104 genes were upregulated, while 4643 genes were downregulated which provide an overview that DS-1 induced change in the transcriptome levels relating to numerous pathway. The specific genes were added to the heatmaps of EMT and adherens junction on the right side. Hope that will be helpful.

Globally check that you use either 24 and 48 h or 24 h and 48 has both are used.

Please check the document so you uniformly use 2.0 µM and 5.0 µM or 2.0 and 5.0 µM both formats are used

Response: Thank you very much for your valuable corrections. The description of 48 h or 24 h was revised to 24 and 48 h in the text. The description of 2.5, 5.0 and 7.5 µM was the finalized style.

Need a brief introduction as to what you are doing what is the reason for this study. (for EMT pathway)

Response: Thank you very much for your critical suggestions. Since EMT is critical to cancer metastasis, the decrease of FASN indicated the hypothesis that DS-1 could inhibit EMT pathway. So that we checked the marker proteins of EMT pathway.

Special response: We would like to express our sincere appreciation to the reviewer for point-to-point marking the comments or correction in the pdf file. It is very useful for improving our manuscript. We also responded to this comments in the reviewer’s pdf file.

General Response: Thank editors and reviewers very much for critical reviews and constructive suggestions as well as detailed corrections. We have tried our best to revise our manuscript carefully according to the reviewer’s and editor’s comments. We sincerely hope that the revised manuscript will meet with approval. Once again, thank you very much for your comments and suggestions. If it is still not satisfactory, please contact us ([email protected], 86-13968131541). We will revise it until it is OK.

Reviewer 3 Report

This study is one of the most meticulously designed and well planned I have seen. Authors comprehensively studied almost all aspects of DS1 in teens of its inhibitory effect of cell migration and enhancing cell attachment capacity, nominating it as a potential anticancer therapeutic naturally-driven drugs. Diverse techniques were imployed that helped authors to come up with a detailed mechanism of action scheme including:

MTT cell proliferation, cell cycle analysis, immuno fluorescence, RNA-Seq, RTPCR, cell migration (wound healing) as well as cell adhesion. My only reservation is that authors need to give more care to the English style as there are several grammatical mistakes and typos scattered through the MS.

I accept the MS after minor revision. Good luck for the authors and I congratulate them for such a well planned and highly novel study. 

Author Response

Comments and Suggestions for Authors

This study is one of the most meticulously designed and well planned I have seen. Authors comprehensively studied almost all aspects of DS1 in teens of its inhibitory effect of cell migration and enhancing cell attachment capacity, nominating it as a potential anticancer therapeutic naturally-driven drugs. Diverse techniques were imployed that helped authors to come up with a detailed mechanism of action scheme including:

MTT cell proliferation, cell cycle analysis, immuno fluorescence, RNA-Seq, RTPCR, cell migration (wound healing) as well as cell adhesion. My only reservation is that authors need to give more care to the English style as there are several grammatical mistakes and typos scattered through the MS.

I accept the MS after minor revision. Good luck for the authors and I congratulate them for such a well planned and highly novel study. 

Response: Thank you very much for the kind comments and positive recommendation. The English style and languages errors have been corrected and whole manuscript was revised in-detail. We hope that the revised manuscript will meet the requirements.

General Response: Thank editors and reviewers very much for critical reviews and constructive suggestions as well as detailed corrections. We have tried our best to revise our manuscript carefully according to the reviewer’s and editor’s comments. We sincerely hope that the revised manuscript will meet with approval. Once again, thank you very much for your comments and suggestions. If it is still not satisfactory, please contact us ([email protected], 86-13968131541). We will revise it until it is OK.

Round 2

Reviewer 1 Report

In the discussion section, the authors have made some significant improvements, but the English is still very poor, even in the newly added parts of the text. Sometimes it is very difficult to understand what the authors want to say.

Therefore, I strongly advise the authors to once again ask an English-speaking colleague for help. The paper cannot be published in the form in which it is presented.

In addition, there are many typos (e.g. line 110: "invioles"), incorrect use of singular or plural (e.g. line 120: "the transcription ... were depressed") or different use of abbreviations (e.g. Ds-4 and Ds-5 instead of DS-4 and DS-5). In the newly added paragraphs, lines 367-373, 383-400, and 448-466, there is an error in almost every sentence. Authors should check their manuscript carefully.

In principle, this is a solid and well-conducted study and the results are worthy of publication. But the presentation needs considerable improvement.

Author Response

Comments and Suggestions for Authors

In the discussion section, the authors have made some significant improvements, but the English is still very poor, even in the newly added parts of the text. Sometimes it is very difficult to understand what the authors want to say. Therefore, I strongly advise the authors to once again ask an English-speaking colleague for help. The paper cannot be published in the form in which it is presented.

Response: Thank you very much for your careful review. As pointed by reviewer, indeed, there are many language errors in previous manuscript although its primary text has been edited by MDPI English pre-edit service (English editing ID: english-35326) and final submitted text has revised by us for many times (at least 10 times) before submission. Thus, in this revision, we have tried our best to improve the manuscript and made some changes to the revised manuscript according to the editor and reviewers’ comments. Now we send the revised manuscript (Text.doc) to you. We have also uploaded a “Text with track changes.doc”. Please find these files and responses. In the “Text with track changes.doc”, all revisions were marked. We hope that the revised manuscript will meet the requirements. If it is still not satisfactory, please contact us. We will revise it until it is OK.

In addition, there are many typos (e.g. line 110: "invioles"), incorrect use of singular or plural (e.g. line 120: "the transcription ... were depressed") or different use of abbreviations (e.g. Ds-4 and Ds-5 instead of DS-4 and DS-5). In the newly added paragraphs, lines 367-373, 383-400, and 448-466, there is an error in almost every sentence. Authors should check their manuscript carefully.

Response: Thank you very much for your careful review and critical comments. They are very useful for improving our manuscript. Indeed, there are many typos and language errors. We revised these errors as possible. Please find these corrections in the “Text with track changes.doc”. We hope that the revised manuscript will meet the requirements.

General Response: Thank editors and reviewers very much for critical reviews and constructive suggestions as well as detailed corrections. We have tried our best to revise our manuscript carefully according to the reviewer’s and editor’s comments. We sincerely hope that the revised manuscript will meet with approval. Once again, thank you very much for your comments and suggestions. If it is still not satisfactory, please contact us ([email protected], 86-13968131541). We will revise it until it is OK.